

**Drone-based multispectral differentiation of subalpine vegetation at the treeline in the**
**Southern Alps of New Zealand**
Fabian Döweler[1*] and Martin Karl-Friedrich Bader[2*]
[1] Dragonfly Data Science, Wellington, New Zealand
[2] System Earth Science, Maastricht University, Brightlands Campus Greenport Venlo, The
Netherlands
*Correspondence:
fab.doe@gmx.de, martin.bader@maastrichtuniversity.nl

**Keywords**
Vegetation index, unoccupied aerial vehicle (UAV), machine learning classifier, mountain
biodiversity
**Abstract**
Subalpine ecosystems are highly dynamic environments that are particularly vulnerable to
environmental change, yet their remote and rugged nature poses challenges for long-term
monitoring. Unoccupied aerial vehicles (UAVs) equipped with multispectral sensors offer a
scalable solution for high-resolution vegetation mapping in these landscapes. In this study,
we integrated UAV-derived spectral data with machine learning (ML) classifiers to assess
the effectiveness of different vegetation indices (VIs) in distinguishing subalpine plant
communities. Principal component analysis (PCA) revealed that NDVI, SIPI2, MCARI, and
CHL were highly correlated and strongly influenced the primary variance in the dataset,
while NDRE and LCI contributed more evenly across principal components, and GNDVI was
largely independent. Among the ML classifiers tested, extreme gradient boosting (XGBoost)
achieved the greatest overall accuracy (81.3%) and Kappa (0.75), outperforming support
vector machines (SVM) and random forest (RF). Classification confidence was highest for
*Chionochloa* tussock (64.6–69.7%) and *Dracophyllum* scrub (70.6%), suggesting moderate
reliability for these dominant vegetation types. Scrub and prostrate mat-forming communities
exhibited lower classification accuracy, likely due to their heterogeneous canopy structure
and greater spectral variability. The ecological boundaries of the subalpine zone, formed by
*Fuscospora* forest and scree, were classified with high confidence, but the vegetation is
dominated by tussock and shrubland. Feature importance analysis ranked NDVI, SIPI2,
CHL, and MCARI highly in SVM and RF models, whereas LCI prevailed in XGBoost,
underscoring how different algorithms leverage spectral information in classification tasks.
These results emphasize the role of vegetation structure in classification accuracy, with
dense, spectrally homogeneous vegetation types more reliably distinguished than mixed-
species communities. Our study highlights UAV-based classification as a valuable tool for
landscape-scale monitoring of subalpine vegetation. As UAV applications and ML workflows
continue to evolve, optimizing classification approaches will enhance our ability to track
ecological changes in subalpine and alpine regions worldwide.



**1. Introduction**

Globally, subalpine shrublands play a crucial role as biodiversity hotspots, supporting a wide
range of endemic plant species and serving as critical habitats for various alpine-adapted
fauna. Functioning as ecological transition zones, they facilitate interactions between
species from lower and higher elevations, leading to unique assemblages. These
ecosystems are also vital for carbon sequestration (Day et al., 2023), water regulation
(Nicholls, 2023; Nicholls & Carey, 2021), and plant-soil nutrient balance (Urbina et al., 2020),
exhibiting key ecosystem services in mountainous landscapes. Additionally, subalpine
vegetation plays a fundamental role in global mountain ecosystems by acting as a crucial
buffer against climate-driven changes (Hou, 2024) and microhabitat facilitation (Harsch et
al., 2009). These environments are highly dynamic, shaped by the interplay of climate,
topography, and ecological processes. However, ongoing climate change is reshaping
subalpine ecosystems worldwide, altering species distributions, ecosystem functions, and
landscape stability (Reid et al., 2022).

The vulnerability of alpine and subalpine vegetation to changing abiotic drivers is particularly
concerning, as many species possess limited dispersal capacities and may be limited in their
capability to respond to changes in their range limitations (Camac et al., 2021). Moreover,
research on climate-driven treeline shifts (Körner, 2014) and microclimatic variability
underscores the complexity and potential of subalpine ecosystems to modulate large scale
abiotic drivers (Döweler et al., 2021, 2024) and biotic effects (e.g. control invasive species
expansion; (Padalia et al., 2023). While some species may benefit from a warming climate,
others, particularly alpine specialists, may not be able to compete with generalist species
expanding their range from lower elevations (Thomas et al., 2023). This ecological
reshuffling has profound implications for biodiversity, carbon storage, and ecosystem
resilience, but often happens gradual and can only be thoroughly studied at the landscape
scale, where large scale assessments of change of the subalpine lacks temporal and spatial
resolution to adequately reflect these changes (Döweler et al., 2024).

In New Zealand, subalpine vegetation is characterized by a mosaic of tussock grasslands
and low-stature shrubs, forming ecologically significant communities that influence
ecosystem resilience and carbon storage (Mark, 2013; Day, 2023). The high cover of
*Chionochloa* tussocks (Fig. 1), along with species such as *Dracophyllum uniflorum*,
*Podocarpus nivalis*, and *Acrothamnus colensoi*, creates microclimates that buffer
temperature extremes and support treeline regeneration of *Leuphozonia menziesii* and
*Fuscospora cliffortioides* (Hook.f.) Oerst. (Döweler, 2021; Scherrer & Körner, 2010). These
ecosystems are not only important for biodiversity but also provide a range of ecological
functions, influencing water retention (van Galen et al., 2023) and soil carbon dynamics (Day
et al., 2023).



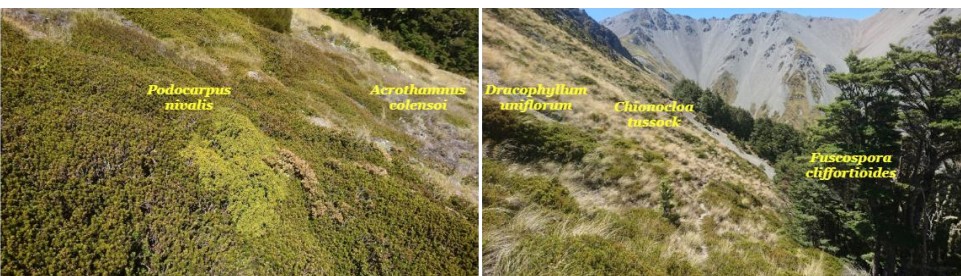


**Figure 1** A typical subalpine belt vegetation composition in New Zealand (1365 m a above
sea level, Craigieburn Valley, Arthurs Pass)
Despite their ecological significance, New Zealand's subalpine landscapes are experiencing
complex transformations, yet our understanding of these changes remains limited. The lack
of landscape-scale detailed vegetation mapping in the subalpine belt limits insights to
capture these gradual but potentially significant shifts in over time (Day et al., 2023) in these
often inaccessible regions. While treelines have shown limited upslope movement in
response to warming (Harsch et al., 2012), subalpine vegetation composition is shifting due
to factors such as woody encroachment and thickening, increased drought stress, and the
potential expansion of invasive species (Chardon et al., 2024; Singh et al., 2024).
Understanding how these communities are responding to climate change is crucial to further
elucidate their role in stabilising these ecosystems to help inform conservation strategies
(Reid, 2022; De Toma, 2025).
Remote sensing is a powerful tool for monitoring subalpine environments, where limited
accessibility and the need for large-scale landscape assessments pose significant
challenges (Walsh et al., 2009). It enables the classification of vegetation and detection of
ecological shifts, offering a comprehensive perspective on mountain biodiversity and
ecosystem dynamics. As climate change increasingly affects alpine and subalpine
ecosystems, the ability to remotely assess vegetation composition across vast and often
inaccessible areas has become a powerful method to study these ecosystems (Garbarino et
al., 2023). Advances in high-resolution satellite and UAV-based remote sensing, combined
with machine learning, have significantly improved vegetation classification, enhancing
mapping accuracy and long-term monitoring (Mashiane et al., 2024; Nguyen et al., 2022).
Access to light-weight sensors which can readily be mounted on increasingly affordable
unoccupied aerial vehicles (UAV's) enables us to monitor subalpine ecotone in
unprecedented detail using optical, multispectral, thermal, and LiDAR sensors (Döweler et
al., 2024). Remote sensing technologies provide a robust means of tracking vegetation
dynamics at ecologically meaningful scales, with satellite and aerial imagery proving
effective in mapping subalpine vegetation and detecting temporal changes (De Toma et al.,
2025). In some field studies, UAV-based deep learning methods may outperform human
observers in delineating complex patterns in subalpine shrub communities (Moritake et al.,
2024), endorsing their use for larger mapping endeavours of the subalpine in an approach to
more accurately study vegetation shifts in response to climate change. These technological
advancements offer critical insights for conservation planning and land management,
ensuring more effective strategies for protecting subalpine ecosystems (Padalia et al., 2023).



This study investigates the potential of remotely sensed optical and multispectral vegetation
indices to differentiate vegetation composition in a complex subalpine shrubland ecotone in
New Zealand. Building on our previous classification and segmentation research in the
Craigieburn Range (Arthur's Pass, New Zealand, Döweler et al., 2024), we aimed to
compare the performance of three widely used machine learning classifiers (support vector
machine, random forest and extreme gradient boosting) and, by extension, to identify which
vegetation indices are most effective for distinguishing vegetation classes. This study aims
to offer recommendations for vegetation indices and ML classifiers for future remote sensing
applications in subalpine ecosystems. We hypothesised that classification accuracy will be
highest in vegetation types with distinct spectral reflectance signatures and relatively low
spectral and structural variability (e.g. sparsely vegetated scree, subalpine forest, tussock)
and lowest in various types of scrub and prostrate mats where increased species
interspersion may cause greater spectral overlap.
**2 Material and Methods**
**2.1 Study site**
The Craigieburn Valley study site (-43.111, 171.713) is located at 1365 metres above sea
level on a southeast to southwest aspect within the eastern slopes of the Southern Alps,
New Zealand (Fig. 2). The site is characterized by a montane to subalpine climate, with
frequent frost events throughout the year (approximately 135 frost days annually) and an
annual rainfall of around 1300 mm. The subalpine belt is dominated by *Chionochloa* tussock
grasslands, interspersed with species such as *Dracophyllum uniflorum* Hook.f., *Podocarpus*
*nivalis* Hook., and *Acrothamnus colensoi* (Hook.f.) Quinn, alongside areas of exposed scree.
The adjacent treeline is formed by *Fuscospora cliffortioides* (Hook.f.) Oerst., marking the
transition to the alpine treeline.



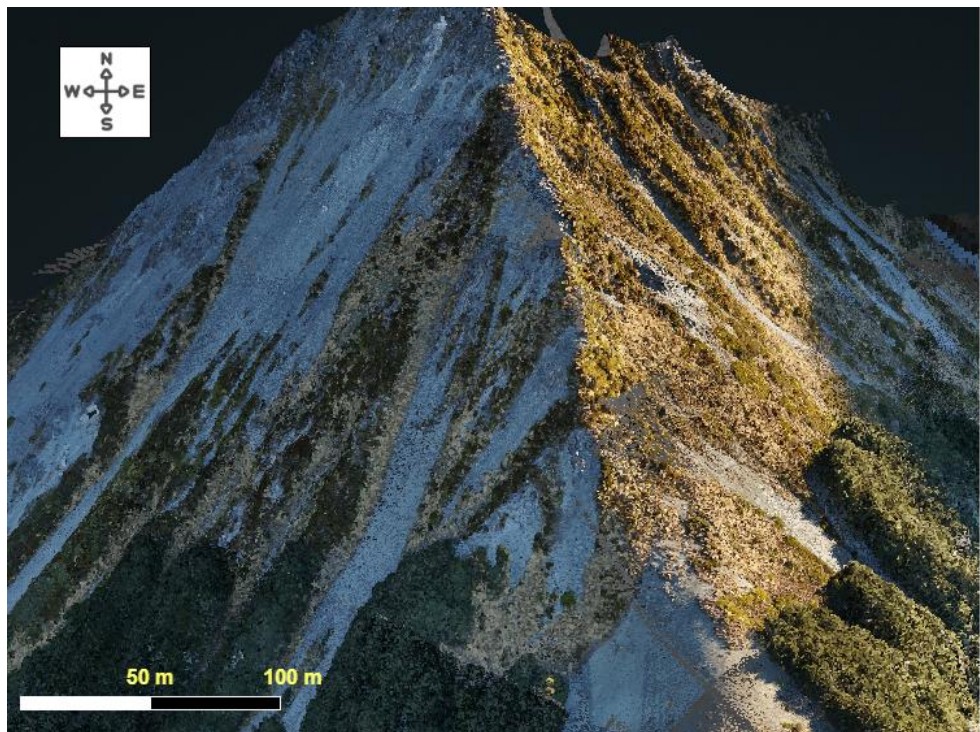

**Figure 2** Craigieburn study site (~ 4 ha). The subalpine belt has been covered by the drone survey. The visualisation shows an RGB coloured point cloud derived from a 2019 (May, 9 am austral winter) point cloud acquisition.

## 2.2 Subalpine vegetation classification and segmentation

For the classification and segmentation of the subalpine belt we we applied the same methodology as previously described in Döweler et al., 2024 producing a detailed map (0.05 m) of the subalpine vegetation, with low spatial offset (6.14 ± 4.03 cm; mean ± standard error) over an area of approximately 4 ha covering the supaline belt. We provide a brief summary of the method below, for the full workflow please see the respective paper.

For georeferencing, we used a differential GPS rover-base setup (Emlid Reach RS+) to locate ground control points (GCPs) consisting of chessboard-patterned panels or high-visibility spray markers, which were distributed across the UAV flight path for photogrammetric orthorectification. The differential GPS was also used to validate classification results by geolocating vegetation patches within the subalpine, resulting in over 600 ground-truth vegetation identifications in Craigieburn Valley. UAV imagery was captured using a DJI Phantom 4 (0.01 m RGB resolution) and a Parrot Sequoia+ multispectral sensor (0.05 m resolution), mounted for aerial surveys during the 2018/19 austral growing season. Pre- and post-flight calibration followed the One-Point Calibration plus Sunshine Sensor method using the Parrot target plate. The UAV flight paths were planned using UgCS software (SPH Engineering, 2025), incorporating an 8 m resolution digital terrain model



(Geographx, 2016) for altitude control, with a 5 m/s flight speed and 80% along-track overlap
for photogrammetric processing.
Post-processing was performed in Pix4D 9.4 (Pix4D, 2025) for image orthorectification and
spatial alignment using mapped GCPs. RGB and multispectral datasets were aligned
manually and projected into the New Zealand Transverse Mercator (NZTM 2000) coordinate
system. Object-based image analysis was conducted using eCognition Developer 9.0
(Trimble, 2025), applying multiresolution segmentation (scale: 250, shape: 0.1,
compactness: 0.7) to generate spectrally homogeneous objects. A nearest neighbor
classifier, trained on geolocated field data, was used to categorize land cover into five
vegetation types and scree. Feature selection was optimized through the Feature Space
Optimization tool in eCognition, maximizing vegetation separability across spectral bands.
Classification accuracy was evaluated using an error confusion matrix based on the 600
ground-truth points, with Kappa statistics indicating an overall accuracy of 89.7% (Fig. 3,
Döweler et al., 2024)

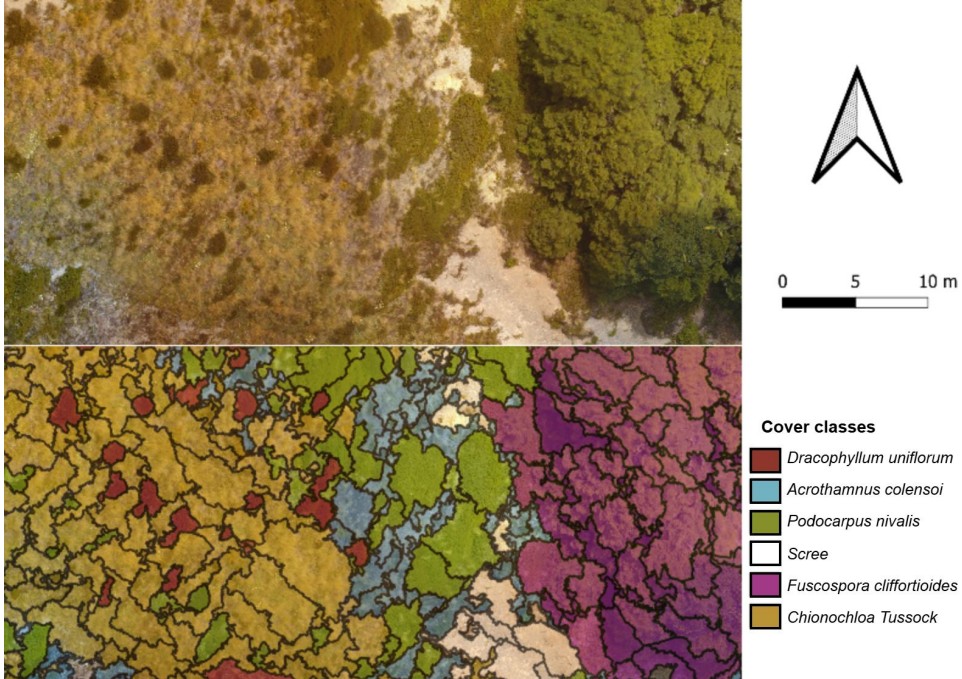

**Figure 3** Results for the pre-existing subalpine vegetation segmentation and classification
for the Craigieburn valley established by Döweler et al. (2024)
**2.3 Vegetation indices used to discern cover classes**
Following the segmentation and classification of the vegetation cover classes, we derived a
suite of vegetation indices from the available multispectral bands to test their capability in
discerning cover classes. These indices capture plant functional traits that influence





productivity, stress responses, and spectral variability across different vegetation types. By
leveraging multispectral reflectance data, we aimed to improve vegetation classification
accuracy and investigate how vegetation properties vary across topographic gradients.
All UAV-derived spectral data was sampled to a uniform 0.01 m resolution using a bicubic
interpolation in GDAL (3.4.1; Rouault et al., 2025). Vegetation classification polygons were
processed in GeoPandas (1.0.1; Van den Bossche et al., 2024) by extracting their centroids,
which were then buffered by 0.2 m to create localized sampling zones. This buffering step
minimized the influence of mixed-pixel effects and ensured that spectral values extracted
within each zone were representative of potential spectral heterogeneity within a single
vegetation type. We used a spatial filtering GeoPandas, ensuring that each buffered centroid
remained entirely within its original vegetation class. After filtering, the final vegetation
sample counts were 4436 *Fuscospora cliffortioides*, 1809 scree, 1384 *Chionochloa* tussock
(-12%), 883 *Acrothamnus colensoi* (-13%), 676 *Podocarpus nivalis* (-10%), and 293
*Dracophyllum* spp. (-3%), with a total of 9476 vegetation samples retained for analysis. The
filtered dataset was used to compute zonal statistics for each vegetation index using the
rasterstats (0.15.0; Perry, 2025) package, extracting median, mean and standard deviation
values within each segmented vegetation type.
The vegetation indices used in this analysis (Table 1) are widely applied in remote sensing
analyses on vegetation health and growth. We included the Normalized Difference
Vegetation Index (NDVI), Structure Insensitive Pigment Index 2 (SIPI2), Modified Chlorophyll
Absorption Ratio Index (MCARI), Green Normalized Difference Vegetation Index (GNDVI),
Chlorophyll Index (CHL), Normalized Difference Red Edge Index (NDRE), and Leaf
Chlorophyll Index (LCI). NDVI was calculated to assess overall vegetation health, while
SIPI2 was included as an indicator of vegetation stress and pigment ratios. MCARI, GNDVI,
CHL, and LCI were shifting the focus on chlorophyll concentrations and photosynthetic
potential, while the NDRE has been selected to provide a focus on red-edge detectable
indication of early stress.
**Table 1** Overview of vegetation indices and respective band calculations used in the current
study, which represent unpublished data from a previous remote sensing study in the same
area (Döweler et al., 2024). Green (550 nm), Red (660 nm), Near-infrared (790 nm, NIR),
Red Edge (735 nm, RE) derived from the Parrot Sequoia, Blue band (450 nm) extracted
from the Phantom 4 RGB sensor.

| Index | Name | Calculation | Reference |
|-------|------|-------------|-----------|
| NDVI | Normalized Difference Vegetation Index | $\dfrac{NIR - Red}{NIR + Red}$ | Rouse et al., 1973 |
| SIPI2 | Structure Insensitive Pigment Index 2 | $\dfrac{NIR - Blue}{NIR - Red}$ | Peñuelas et al., 1995 |
| MCARI | Modified Chlorophyll Absorption Ratio Index | $[(RE - Red) - 0.2 \times (Red\ Edge - Green)] \times \dfrac{RE}{Red}$ | Daughtry et al., 2000 |



| Index | Name | Calculation | Reference |
|-------|------|-------------|-----------|
| GNDVI | Green Normalized Difference Vegetation Index | $$\frac{NIR - Green}{NIR + Green}$$ | Gitelson & Merzlyak, 1998 |
| CHL | Chlorophyll Red-Edge Index | $$\frac{NIR}{RE} - 1$$ | Gitelson et al., 2003 |
| NDRE | Normalized Difference Red Edge Index | $$\frac{NIR - RE}{NIR + RE}$$ | Barnes et al., 2000 |
| LCI | Leaf Chlorophyll Index | $$\frac{NIR - RE}{NIR + Red}$$ | Haboudane et al., 2002 |

**2.4 Statistical analysis**
All statistical computations and graphics were performed using the R software within the
RStudio integrated development environment (R version 4.4.1, R Core Team, 2024, RStudio
version 2024.09.0+375, Posit team, 2024). To provide a general overview, we performed a
principal component analysis (PCA) based on the overall medians of the vegetation indices
(i.e. the median of the sample medians of each index). The PCA results were visualised in a
biplot to illustrate similarities among vegetation types and the contribution of VIs to the
principal components (loading vectors) as well as their interrelationships.
We used three popular machine learning approaches to classify the six vegetation types
based on the centroid buffered medians of seven features (vegetation indices): NDVI,
GNDVI, CHL, LCI, MCARI, NDRE, SIPI2. We created a balanced 80/20 training-to-test split
of our data, ensuring random sampling within each class to preserve the overall class
distribution (balanced splits are obtained by providing a factor, i.e. the vegetation type labels,
to the *createDataPartition* function, R package *caret*, Kuhn, 2008)
We trained a Support Vector Machine (SVM) classifier using a radial basis function (RBF)
kernel to differentiate between vegetation types (R package *e1071*, Meyer et al., 2024)
To address class imbalances, we assigned class weights inversely proportional to their
frequencies (following the approach in the Python package *scikit-learn*, Pedregosa et al.,
2011). Hyperparameter optimization was performed via 5-fold cross-validation, selecting the
optimal values of C (regularization parameter) and γ (kernel coefficient, controlling the
influence of data points) using a grid search. The final model featured an RBF kernel with C
= 10 and γ = 0.01.
We also ran a random forest approach to classify the vegetation types (R package
*randomForest*, Liaw & Wiener, 2002). The model algorithm was run with the default 500
number of trees and a hyperparameter tuning procedure suggested two randomly sampled
features (predictors) at each split (mtry = 2). Stratified sampling was used to ensure that
each tree was trained on a random sample containing observations from all vegetation
types.
In addition, we applied extreme gradient boosting (function *xgb.train* in R package *xgboost*,
Chen et al., 2024) (Chen et al., 2024) for vegetation type classification. A grid search was
used for hyperparameter tuning (final hyperparameter settings: eta = 0.05, max_depth = 6,
gamma = 2) and 5-fold cross-validation to determine the optimal number of iterations (100
cross-validation runs allowing a maximum of 500 iterations yielded a mean of 150 iterations).
For all three machine learning approaches, the model performance scores were derived
from a confusion matrix contrasting true and predicted class labels (*confusionMatrix* function



in R package *caret*, Kuhn, 2008). Class-level sensitivity (recall) and specificity scores were
each averaged to an overall score. In addition, the multi-class area under the receiver
operating characteristic curve ($AUC_{mc}$) was calculated as the mean of the class-specific
AUCs (function *multiclass.roc* in R package *pROC*, Robin et al., 2011). Finally, we applied a
permutation-based feature importance analysis to all three classifiers. This feature
importance procedure was run with 30 permutation rounds using a cross-entropy loss
function to evaluate feature contribution (*feature_importance* function in R package
*ingredients* relying on the cross-entropy loss function in R package *DALEX*, Biecek, 2018;
Biecek et al., 2023).
**3 Results**
**3.1 Principal component analysis (PCA)**
The PCA showed that PC1 explained roughly 95 % and PC2 nearly 5 % of the variation in
the aggregated data. In the PCA biplot, *Podocarpus* scrub and *Dracophyllum* scrub formed
the only discernible cluster (which could perhaps include *Chionochloa* tussock), indicating
similar profiles related to the VIs. By contrast, the remaining vegetation types showed
distinct profiles (Fig. 4). The small angular distances between the loading vectors of NDVI,
SIPI2, MCARI and CHL suggest that these indices were all positively correlated and their
fairly horizontal alignment indicates a strong influence on PC1 (Fig. 4). The LCI and NDRE
were also positively correlated and contributed roughly equally to both PCs. The GNDVI had
a strong negative contribution to PC2, and was weakly or uncorrelated with the other VIs
considering the large angular distances to the other loading vectors (Fig. 4).

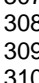
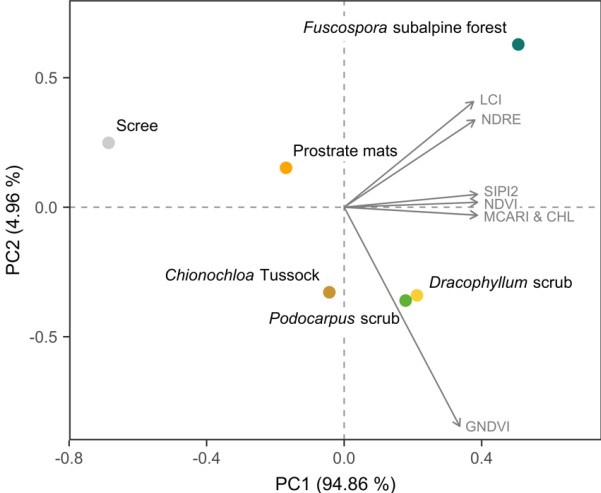

**Figure 4** Principal component analysis (PCA) biplot displaying the scores associated with various
subalpine vegetation types (filled circles) and the contribution of vegetation indices (arrows indicating
loading vectors) to the principal components. The vegetation indices were scaled prior to PCA. Please
note that the eigenvectors of the MCARI and CHL indices are virtually equal.
**3.2 Machine learning classifiers**



Based on common performance metrics (accuracy, Cohen's Kappa statistic, AUC, sensitivity
and specificity), the support vector machine (SVM) and random forest (RF) classifiers
performed similarly well but were slightly outperformed by the extreme gradient boosting
algorithm (XGBoost) (Fig. 5). Accuracy was just below 80 % in the SVM and the RF, while
reaching 81.3 % in the XGBoost approach. Given our unbalanced data, the more robust
Kappa statistic seems more informative than overall accuracy. Cohen's Kappa was highest
in the XGBoost classifier at 0.75, compared to 0.73 in the SVM and 0.72 in the RF approach
(Fig. 5). The multi-class AUC-ROC and specificity were identical in all three algorithms
($AUC_{mc}$ = 0.93, specificity = 0.96), while sensitivity (recall) varied from 0.72 in the SVM and
0.70 in the XGBoost to 0.68 in the RF classifier (Fig. 5). As judged by the percentage of
correct classifications in the confusion matrices of the three classifiers, *Fuscospora*
subalpine forest (93.3 – 94.7 %) and scree (86.7 – 88.7 %) can be identified with high to very
high confidence, followed by moderate classification confidence for *Chionochloa* Tussock
(64.6 – 69.7 %). Classification confidence for the remaining vegetation types was mostly low
(< 60 %), except for the SVM's 70.6 % correct classifications of *Dracophyllum* scrub and its
63.2 % accuracy in classifying *Acrothamnus colensoi* (Fig. 5).



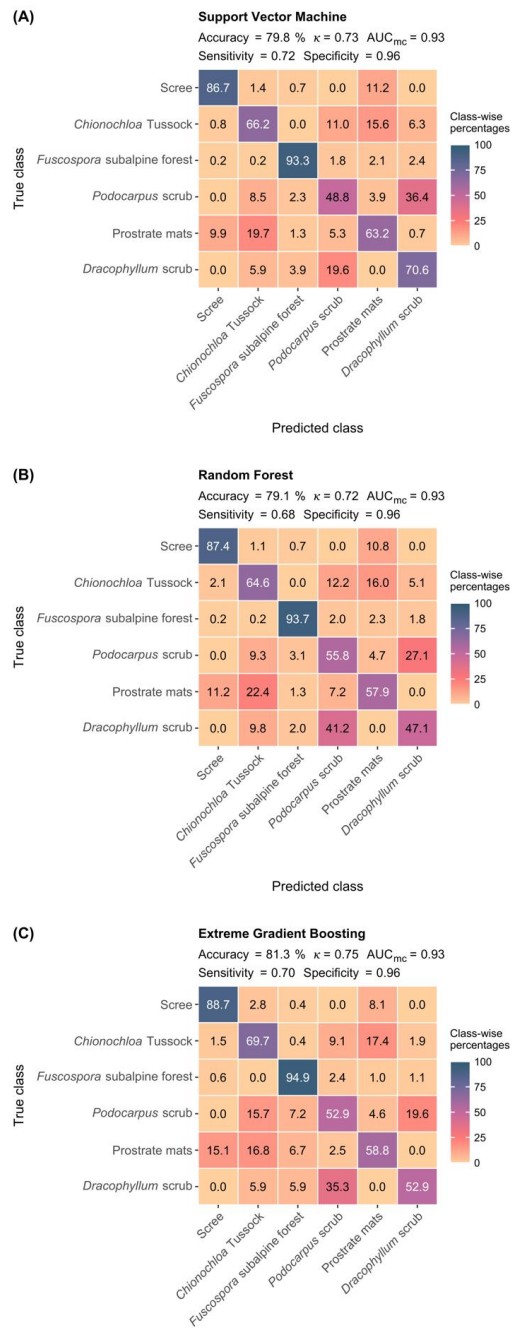

**Figure 5** Heatmaps of the confusion matrices of a support vector machine (**A**) a random forest model (**B**) and an extreme gradient boosting machine (**C**). $\kappa$ indicates Cohen's Kappa statistic, $AUC_{mc}$ denotes the multi-class area under the ROC curve. Accuracy, $\kappa$ and $AUC_{mc}$ are overall model statistics, while sensitivity (recall) and specificity indicate averages of the class-specific metrics.



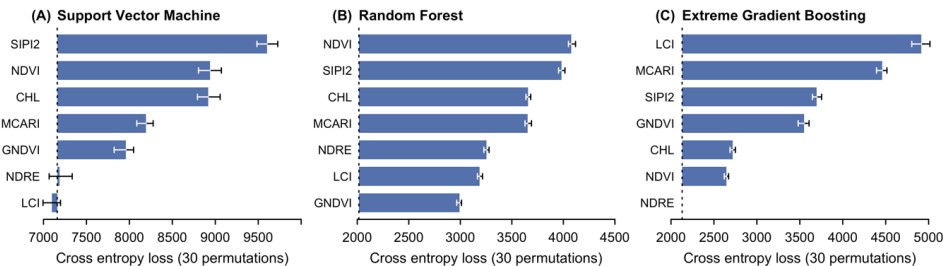

**Figure 6** Permutation-based feature importance (variable importance) of a support vector machine (**A**), a
random forest model (**B**), and an extreme gradient boosting model (**C**). The vertical dotted lines indicate
the cross-entropy loss of the full model (no variables dropped). The error bars signify the 2.5th and 97.5th
percentiles of the 30 permutations.
In the SVM algorithm, SIPI2 represented the most important variable, followed by the equally
relevant NDVI and CHL, while MCARI and GNDVI were of moderate importance (Fig. 6 A).
NDRE and LCI had little to no significance in the SVM. In the RF model, NDVI and SIPI2
were similarly important, followed by equal contributions from CHL and MCARI, and to a
lesser but similar extent from NDRE and LCI, with GNDVI coming in with a slightly lower
entropy loss value (Fig. 6 B). By contrast, the XGBoost algorithm relied most heavily on LCI,
followed by MCARI and, on a similar level, SIPI2 and GNDIV (Fig. 6 C). CHL and NDVI were
of lesser importance, while NDRE represented an entropy-irrelevant feature.
**4. Discussion**
Mountain ecosystems support a high level of plant diversity and endemism, especially at
high elevation but coordinated monitoring efforts are scarce (Perrigo et al., 2020), often
hampered by limited access. In the future, more affordable drone technology will bolster
remote sensing based mapping and monitoring of these hard-to-reach ecosystems, and
information that aids in optimising classification accuracy will facilitate progress in this field.
Here, we compared the performance of three popular ML classifiers and assessed their
feature importance to rank common VIs based on their impact on the classification algorithm.
Judged by the overall accuracy and Cohen's Kappa, the XGBoost algorithm differentiated
the existing vegetation types best, but SVM and RF both performed nearly as well.
Consistent with our findings, XGBoost outperformed SVM and RF in urban land use-land
cover and forest classification tasks (Georganos et al., 2018; Ramdani & Furqon, 2022). In
line with our hypothesis, spectrally more distinct vegetation types with lower spectral and
structural variation showed the greatest proportion of correct classifications in the confusion
matrices of the three classifiers, which is consistent with previous findings reported for
diverse wetland ecosystems (Schmidt & Skidmore, 2003). In our study, these spectrally
distinct vegetation types included *Fuscospora cliffortioides* subalpine forest and *Chionochloa*
tussock, which are characterised by dense, uniform foliage and/or well-defined canopy
structures, making them easier to differentiate in the ML classification process (Ollinger,
2011). The sparsely vegetated scree is characterised by a relatively uniform spectral
signature, i.e. spectrally bland, which facilitates classification.
The importance of the used VIs was ranked similarly in the SVM and RF but differed greatly
in the XGBoost algorithm, suggesting that the different ML approaches rely on distinct
spectral properties for classification. The SVM and the RF model agreed closely in regard to



the four most important VIs (SIPI2, NDVI, CHL, MCARI). These VIs reflect pigment content
and general vegetation vigor and are closely related to photosynthetic activity, which makes
them suitable for distinguishing between broad vegetation types. By contrast, in the XGBoost
approach LCI emerged as the most important feature, which had minimal leverage in the
other two classifiers. This discrepancy likely reflects XGBoost's greedy search for splits that
minimise the loss function, allowing a feature to gain importance even if it is of little
relevance in SVM and RF algorithms (Kamdem and Fokoue, 2022). LCI's top ranking in
XGBoost suggests it may hold critical information for distinguishing subtle variations in leaf
structure or pigment content that were not prioritized in the RF and SVM algorithms.
A recent review on plant and vegetation classification based on spectral signatures revealed
that besides the biological properties also the methodological approaches, the scale at which
the recordings are performed and not least the applied feature selection procedure itself may
all have a strong influence on feature importance in ML classifiers (Hennessy et al., 2020).
To eleminate the latter source of variation, we applied the same permutation-based feature
importance analysis based on cross-entropy loss to all three classifiers (Biecek, 2018;
Biecek and Baniecki, 2023).
Another notable finding is the weak contribution of NDRE in all classifiers, indicating its
limited role in distinguishing subalpine vegetation types for the subalpine ecotone at our
study site, which calls for verification in other high-elevation transition zones. Unlike NDVI
and MCARI, which are widely used in vegetation classification, NDRE is often associated
with deeper canopy penetration and is particularly useful in detecting nitrogen stress and
subtle variations in chlorophyll content (Boiarskii & Hasegawa, 2019). The lack of importance
of NDRE in this study suggests that these characteristics were not primary drivers of spectral
separability among the subalpine vegetation types analyzed here. Instead, indices related to
general canopy structure and pigment concentration (such as NDVI, CHL, and MCARI)
proved more effective.
The results of our study highlight the critical role of vegetation structure in classification
accuracy, with dense, spectrally uniform vegetation types being more reliably identified than
structurally diverse shrublands and mixed-species communities. As UAV technology
becomes increasingly accessible, further refinement of vegetation index selection and
classification methodologies is essential to capture the often subtle responses of subalpine
vegetation to abiotic stressors, which are being exacerbated by climate change. The decline
of certain species could lead to the loss of critical microhabitats and climatic niches, which
serve as stepping stones for the recruitment of subalpine specialists and treeline forming
species (Döweler et al., 2021; Frei et al., 2018; Harsch et al., 2009). For New Zealand, a
landscape-scale classification of the subalpine can support monitoring the impact of invasive
herbivores on these ecosystems, as their grazing pressure threatens both vegetation
dynamics and the region's carbon sequestration potential (Lee et al., 2000). Advancements
in remote sensing and machine learning offer novel pathways to improve monitoring efforts,
enabling us to more clearly formulate and track conservation targets.
**5. Conclusion**
The effective integration of vegetation indices with modern ML classifiers presents a
powerful tool for tracking ecological shifts, particularly in remote and rugged environments.
As UAV operations become more affordable, their application in long-term monitoring will be
invaluable for detecting and understanding vegetation changes in otherwise inaccessible
regions. Expanding these efforts through global collaboration will provide deeper insights
into the poorly understood dynamics of subalpine ecosystems under changing climatic
conditions. Given the crucial role of subalpine grasslands and woody vegetation in carbon
sequestration (Ward et al., 2014) and other ecosystem services such as maintaining
biodiversity, erosion protection, runoff regulation and snow retention, their ecological



trajectories must be closely monitored to inform conservation strategies aimed at mitigating
species loss and preserving ecosystem functions.
**Data availability**
All data for this publication can be requested from the corresponding author.
**Author contributions**
FD contributed to data acquisition and sample collection and data preparation. MB
contributed to data analysis. Both authors contributed equally to the writing of this
manuscript.
**Competing interests**
The contact author has declared that none of the authors has any competing interests.
**Acknowledgements**
Support for this research was provided by Auckland University of Technology (AUT,
Auckland). We would like to thank the Department of Conservation (DOC) for access
permissions, as well as Lincoln University (Canterbury, New Zealand) for providing
accommodation.

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
