# Peer review of "Drone-based multispectral differentiation of subalpine vegetation at the treeline in the"

_EGUsphere, 2025_

## Author Comment (AC1)

RC1 Comments

Dear Döweler and Bader,

Thank you for your submission. Your study presents an interesting approach that will likely contribute to the scientific community by demonstrating the effectiveness of combining UAV-derived multispectral data with different machine learning classifiers for vegetation mapping.

However, several important revisions are necessary before publication. Below, I outline some general comments followed by more specific suggestions.

Scientific contribution issue: Your work builds upon an existing dataset recently published, with the only modification being the spectral data employed in the analysis. Although this is not inherently problematic, as a study focused on methodology, it is expected that some innovative aspect be clearly presented. The methodological contribution appears limited, as it does not introduce substantial novelty.

We appreciate this concern. While the spatial data source overlaps with our previous work, the current study addresses a distinct methodological question: the ability of UAV-derived vegetation indices, in conjunction with ML classifiers, to discriminate closely related vegetation types in a topographically complex subalpine ecotone. This is novel in both scope and purpose. In a revision, we would clarify this distinction more explicitly in the introduction and methods, and highlight how the comparative evaluation of classifier behaviour (e.g., differing feature importance across models) constitutes a methodological contribution that extends beyond our prior segmentation work.

Scale of analysis: While you provide evidence that the applied methodology is effective at the study site, one of the key advantages of remote sensing is its capability to analyze vast spatial extents. In this work, however, the spatial extent is limited (4 hectares), which makes the term "landscape" analysis somewhat debatable. I suggest either expanding the analysis within the same site or incorporating data from additional sites to improve the study's relevance and achieve a true landscape-scale analysis.

We respectfully disagree that "landscape-scale" should be replaced in this context. While the physical extent of the surveyed area is 4 ha, it captures the full elevational gradient of the subalpine ecotone at the site, a zone that is itself limited in altitudinal extent across the Southern Alps. Moreover, the steep, remote terrain imposes practical constraints on data acquisition, making UAV workflows essential for landscape-level inference. The term "landscape-scale" here refers not only to spatial size but to ecological resolution and representativeness. In a revision, we would

clarify this framing and add language explaining how our workflow is modular and can be applied to larger areas, even if ground validation is only feasible in selected subplots due to accessibility constraints.

If such an expansion is not feasible, I recommend incorporating ecological analyses to assign a more specific aim to your work. For instance, as mentioned on line 434, the output of the machine learning classifiers could be used to monitor and track ecological shifts. Alternatively, as noted in lines 419 to 421, combining vegetation indices with the trained classifiers to investigate the effects of specific abiotic stressors on subalpine vegetation could also provide significant added value. Including an analysis along these lines would enhance the value and appeal of your study, by providing the audience with a concrete example of how the proposed methodology can be applied to address well-known ecological issues and tasks.

This is a valuable suggestion. In a revised manuscript, we propose to elaborate on how classifier outputs could serve as spatial baselines for long-term monitoring, particularly in relation to known abiotic gradients (e.g., drought exposure or elevation). While our current analysis was focused on methodological evaluation, we would expand the discussion to include a conceptual link between UAV-derived vegetation structure and ecological stress monitoring. We will also cite recent works that have implemented this approach in alpine contexts.

Model evaluation: Additional concerns pertain to the evaluation of your models. The manuscript does not clearly explain how the data were partitioned into training, validation, and testing sets. It appears that 80% of the data were used for training and 20% for testing, but how the validation process was performed during model training remains unspecified. More detailed and straightforward information regarding the cross-validation procedure is needed to ensure reproducibility. Lastly, the discussion section cites only a limited number of relevant studies employing XGBoost, SVM, and RF for similar tasks. A more comprehensive comparison with previous applications in analogous contexts would strengthen your results and findings.

In the original submission, we stated clearly the 80/20 split into training and test data: 'We created a balanced 80/20 training-to-test split of our data, ensuring random sampling within each class to preserve the overall class distribution (balanced splits are obtained by providing a factor, i.e. the vegetation type labels, to the *createDataPartition* function, R package *caret*, Kuhn, 2008)'

If this description of the data split is not clear enough, we are happy to rephrase this sentence in our revision. Also, in the revised version of the manuscript, we will

elaborate in more detail on model validation during training, including the used cross-validation procedure.

We agree in terms of relevant references related to the classifiers and will expand the literature review in the introduction and discussion to situate our work in the context of other UAV-based and subalpine vegetation classification studies using these algorithms. We can include recent applications in alpine ecosystems and forest structure monitoring that used XGBoost in particular.

Manuscript template: Please also ensure that your manuscript adheres to the Biogeosciences publication template. The current preprint does not follow the required formatting guidelines—this includes aspects such as title, main text, chapter and subchapter fonts, spacing between paragraphs, figure and table titles and descriptions, and the reference format (Copernicus Publication style).

We will revise the manuscript to fully conform to Copernicus formatting guidelines, including consistent heading styles, figure and table formatting, and the reference style.

Additional suggestions:

Introduction: include a dedicated paragraph providing a brief explanation of the machine learning classifier mentioned at line 133 to give readers essential context.

We will incorporate such a paragraph in the revised version of our manuscript.

Conclusions: The current conclusions are not directly connected to the analyses conducted in the paper. Instead, they focus on reiterating well-known facts already mentioned in the introduction and discussion sections that do not need further discussion. It would be more effective to succinctly summarize the main findings, offering valuable insights into the topic, highlight any key limitations of the applied methodology, clearly emphasize the study's contribution to the scientific community, and propose perspectives and suggestions for future research.

Agreed, we will rephrase the conclusion section accordingly.

Please find below more specific comments and revisions:

L 34-35: "likely due to their heterogeneous canopy structure and greater spectral variability". This fact is reported several times throughout the paper but there is a lack of citations supporting this hypothesis. Please cite other studies underscoring the issue if possible.

Thanks for pointing out this issue. We have found two highly cited references in support of this statement, which we will include in the revised version of our manuscript.

Ollinger SV (2011). Sources of variability in canopy reflectance and the convergent properties of plants. New Phytologist, 189(2):375-94.

Asner GP (1998). Biophysical and biochemical sources of variability in canopy reflectance. Remote sensing of Environment, 64(3):234-53.

L 40: "These results emphasize the role of vegetation structure in classification accuracy"

This is not correct. Your results emphasize the role of the analyzed VIs in classification accuracy. The role of vegetation structure was not explicitly tested and remains an assumption made by the authors. The correlation between vegetation structure and classification accuracy was not investigated in this study. If this assumption is based on findings from other research, please cite those studies and clarify that your classification results could potentially be influenced by vegetation structure.

We acknowledge this and would rephrase these statements to reflect that spectral variation may correlate with structural heterogeneity but was not directly quantified in our analysis. We will also cite studies supporting this interpretation.

L 42-43: "Our study highlights UAV-based classification as a valuable tool for landscape-scale monitoring of subalpine vegetation". This sentence must be changed. This study highlights UAV-based classification as a valuable tool for landscape-scale mapping of subalpine vegetation. Which can later on be used to monitor subalpine vegetation over time. However, since no monitoring was performed in the present paper and there is no evidence it can be done effectively, please modify this statement.

We agree and would revise the statement to say our study demonstrates a UAV-based classification approach that can support future monitoring efforts, while emphasising that our analysis represents a single time point.

Furthermore, your study area consists of a surface of 4 hectares, which is a quite limited extent to be referred to as a "landscape-scale" analysis. A 200m x 200m surface is most probably not representative of the heterogeneity of the landscape in which it is located. Hence, I would rather refer to this analysis with the term "local-scale".

Answered above.

L 79-88: This paragraph looks much like a site description especially if presented along with a figure (Figure 1). As such, both the paragraph and figure could probably fit better in chapter 2.1 (study site).

We agree that it would improve the logical flow and will move this section to the Study Site chapter in a revision.

L 91: "1365 m a above sea level, Craigieburn Valley, Arthurs Pass" remove "a"

L 96: "…but potentially significant shifts in over time" remove "in"

L 97: "inaccessible" modify with "hard to access" or something similar, it is not completely impossible to access the areas.

L 107-109: please add citations of works performing such analyses

L 109-110: "As climate change increasingly affects alpine and subalpine ecosystems" Fact already mentioned before, stick to the remote sensing topic. Ok, acknowledged.

L 117: Please provide more citations Will be included.

L 118-121: Please provide more citations Will be included.

L 121-124: Complex sentence, please simplify for an easier and smoother reading Will be rephrased more clearly.

L 123: add "zone" or "area" after subalpine Will be done.

Chapter 2.1: Provide a picture which gives a more detailed overview of the study site and its features.
i.e. the 4 ha orthomosaic of the study site with a zoom-in of a relevant area in an inset. Will be included.

At the moment the reader does not have a clear idea of the context in which the analysis was performed, the heterogeneity of the topography, vegetation, slope etc.

More relevant information will be provided in the revised manuscript.

Figure 2.1: Try changing compass color to white and remove background color Will be done.

L 165: remove repetition "we" Will be done.

L 238: increase font size of column "reference" in the table Will be done.

L 252-254: not necessary. This is a basic description of the PCA which is also explained throughout next chapters. Ok, will be removed.

L 258: how the training, validation, and test dataset were generated has to be better explained. It is not clear whether 20% of the dataset was used to validate the performances during the training process, or if it was actually an independent test dataset. Please clarify. Will be done.

L 315: "The vegetation indices were scaled prior to PCA". Please specify how they were scaled for reproducibility We will specify this in more detail.

L 374: The citations provided are not sound with the analysis performed in the paper. If possible, please provide citations of more relevant paper conducted in similar contexts and on similar classes. We will strive to do so.

L 386: "suggesting that the different ML approaches rely on distinct spectral properties for classification". Are there any other studies supporting this theory? Please cite them Such references will be included.

L 390: please cite a paper where this operation was performed Will be done.

L 401: spelling error: eliminate Will be corrected.

L 416-418: saying "critical role of vegetation structure in classification accuracy" This was never proven. It should be better to say "the critical role of the spectral information". As it is also mentioned in Chapter 2.3 (L205-208) "we derived a suite of vegetation indices from the available multispectral bands to test their capability in discerning cover classes. These indices capture plant functional traits that influence productivity, stress responses, and spectral variability across different vegetation types", vegetation structure is not directly captured by any of the VIs employed in the analysis. The effect of the structure on the classification was hypothesized, but not proved in the paper, and no reference was cited to support the hypothesis. Agreed, we rephrase things accordingly.

L 419-421: In the present paper the possibility to investigate the effect that an abiotic stressor can have on subalpine vegetation thanks to a specific VI was never tested. Please provide citations of works where this type of investigation was done. Will be included.

L 434: "… tracking ecological shifts". Please provide citations of works doing this analysis. Will be included.

I hope these suggestions are helpful as you revise your manuscript.

We thank the anonymous reviewer for the insightful comments and suggestions, which will definitely help improve our manuscript.

---

## Author Comment (AC2)

RC2 Comments

Dear authors,

your study presents a promising approach to the integration of UAV-derived multispectral data with various machine learning classifiers for vegetation mapping. However, several significant revisions are necessary before your manuscript can be considered for publication, it requires significant changes before it can be resubmitted for further review as a new manuscript. Below, you can find my general comments.

Your published dataset of the previous paper (*https://www.mdpi.com/2072-4292/16/5/840*) was partially used as "ground truth" for a different spectral data in the analysis. While the reuse of your data could be understandable, this manuscript does not introduce substantial new techniques or research novelty. This is highlighted also from the obvious findings, like that the "*spectrally more distinct vegetation types with lower spectral and structural variation showed the greatest proportion of correct classifications in the confusion matrices of the three classifiers*", where the best results are obtained with *Fuscospora* dense forest and sparsely vegetated scree.

We understand this concern and will clearly state how the current analysis differs from the prior study. This manuscript shifts focus from object-based classification to spectral index-based ML classification. Importantly, we analyse feature importance across three classifiers and investigate model performance differences for fine-scale vegetation classes—a novel contribution within this landscape context. We will clarify this distinction and cite our previous work explicitly in the introduction and methods

I recommend a substantial revision of the abstract, as it currently lacks key information regarding the validation or ground-truth data used, the characteristics and spatial extent of the study area, as well as the data collection date. The structure and order of the content in the abstract need to be completely revised. It is widely acknowledged that very few treelines worldwide are entirely unaffected by human influence. This should be acknowledged in the introduction, with consideration given to how such anthropogenic factors—alongside climate—should be incorporated into the modelling framework.

We agree and will revise the abstract to include the 4 ha area, 2018/2019 flight season, use of prior field-based segmentation for ground truth, and summary accuracy/Kappa values. This is an important point, and we will incorporate this into the introduction by noting that while the study area is remote, historical land use

(e.g., grazing) or invasive species pressure may shape vegetation dynamics. We will also discuss how such factors could be integrated into future modelling.

Although it is evident that previously published data were used as ground-truth, the manuscript does not clearly explain how these data were incorporated into the training and/or validation phases. While 600 ground control points are mentioned—points that appear to be notably unbalanced in terms of vegetation cover—it remains unclear how exactly these points were utilized in the analysis. Moreover, it is unclear which steps are part of the previous work and which are new (lines 185-196 and lines 212-225). I recommend that the authors revise this section to enhance clarity and prevent any confusion for the reader. Although reference is made to a previously published paper, it would be advisable to clearly specify in the M&M section the spectral bands available from the multispectral sensor mounted on the UAV.

In a revision, we would explicitly delineate which steps were reused (segmentation, field data acquisition) and which were novel (index derivation, ML classifier training). The 600 field GPS points were used both to validate the original segmentation and to label training polygons for ML classification in this study. This distinction will be made clearer in the methods section.

In your previous study, you classified two alpine treeline ecotones in the Canterbury region of New Zealand's South Island with similar vegetation type. Why not use one site for training and one for model testing and/or validation?

We appreciate the suggestion to use one site for training and another for validation. However, the two ecotones—Craigieburn and Lewis Pass—differ substantially in climatic regime and vegetation composition, despite their geographic proximity. Craigieburn is more arid, with cold winters and well-defined snow cover, while Lewis Pass is shaped by higher summer rainfall and supports different dominant vegetation types. These environmental and floristic differences introduce confounding effects that would compromise model transferability without further ecological normalisation. For this reason, we chose to focus on a single ecotone with full field validation and consistent ecological conditions to evaluate classifier performance.

The multivariate analysis could be reduced/deleted to allow more space for the machine learning classifiers, especially since the PCA does not provide much informative value, with 95% of the variance explained by PC1, and possible strong autocorrelations between many of the spectral indices used. In paragraph 3.2, a very expected result of correct classification for the *Fuscospora* forest is shown, and other class-wise percentages are briefly commented on. The sentence "*Classification confidence for the remaining vegetation types was mostly low (< 60%)*" could be

further investigated, as the challenges might be to distinguish between similar vegetation types. The most significant part of the paper appears to be the discussion of different algorithms applies (Figure 6 and lines 351–358); however, the manuscript lacks a discussion on how the different approaches or algorithms could be integrated, especially given their distinct and complementary behaviors.

We agree that the PCA provides limited explanatory value in its current form, given the dominance of the first component and the intercorrelation of indices. In a revision, we would condense or relocate this analysis to supplementary material to focus more clearly on the machine learning results. Regarding class-wise performance, we acknowledge that lower accuracy in some vegetation types likely reflects overlapping spectral responses in structurally heterogeneous classes such as scrub and mat-forming vegetation. We will expand our discussion of these results and propose that future efforts could incorporate ensemble models or classifier stacking to leverage the complementary strengths observed across SVM, RF, and XGBoost, as their differing feature priorities suggest integration could improve robustness.

Finally, some aspects mentioned in the discussion seem off-topic: e.g. "a landscape-scale classification of the subalpine can support monitoring the impact of invasive herbivores on these ecosystems, as their grazing pressure threatens both vegetation dynamics and the region's carbon sequestration potential", especially considering that the spatial extent of this work cannot be considered sufficient for a 'landscape-scale' study. I suggest modifying the discussion and conclusion sections after a thorough revision of the article.

We acknowledge the reviewer's concern regarding the spatial extent of the study and agree that care must be taken in using the term "landscape-scale." However, we consider it justified here given the full altitudinal coverage of the subalpine belt at the site, the high spatial resolution of the UAV data, and the limited accessibility of these environments. Our use of the term reflects ecological representativeness rather than absolute area. Alternatively, we could use a more specific term like 'ecotone-scale'. That said, we agree that references to herbivore impacts and carbon dynamics are beyond the scope of this analysis and will revise the discussion to limit conclusions to findings directly supported by the data, while moving broader applications to future work.